

# Geostrophic Currents in the northern Nordic Seas from a Combination of Multi-Mission Satellite Altimetry and Ocean Modeling

Felix L. Müller[1], Denise Dettmering[1], Claudia Wekerle[2], Christian Schwatke[1], Marcello Passaro[1], Wolfgang Bosch[1], and Florian Seitz[1]

[1]Deutsches Geodätisches Forschungsinstitut, Technische Universität München, Arcisstraße 21, 80333 Munich, Germany
[2]Climate Dynamics, Alfred Wegener Institute, Helmholtz Centre for Polar and Marine Research, Bussestraße 24, 27570 Bremerhaven, Germany

**Correspondence:** Felix L. Müller (felix-lucian.mueller@tum.de)

**Abstract.**

A deeper knowledge about geostrophic ocean surface currents in the northern Nordic Seas supports the understanding of ocean dynamics in an area affected by sea ice and rapidly changing environmental conditions. Monitoring these areas by satellite altimetry results in a fragmented and irregularly distributed data sampling and prevents the computation of homogeneous and highly resolved spatio-temporal datasets. In order to overcome this problem, an ocean model is used to fill in data when altimetry observations are missing.

The present study provides a novel dataset based on a combination of along-track satellite altimetry derived dynamic ocean topography (DOT) elevations and simulated differential water heights (DWH) from the Finite Element Sea ice Ocean Model (FESOM). This innovative dataset differs from classical assimilation methods because it substitutes altimetry data with the model output, when altimetry fails or is not available.

The combination approach is mainly based on a Principal Component Analysis (PCA) after reducing both quantities by their constant and seasonal signals. In the main step, the most dominant spatial patterns of the modeled differential water heights as provided by the PCA are linked with the temporal variability of the estimated DOT from altimetry by performing a Principal Component Synthesis (PCS). After the combination, the by altimetry obtained annual signal and a constant offset are re-added in order to reference the final data product to the altimetry height level. Surface currents are computed by applying the geostrophic flow equations to the combined topography. The resulting final product is characterized by the spatial resolution of the ocean model around 1 km and the temporal variability of the altimetry along-track derived DOT heights.

The combined DOT is compared to an independent DOT product resulting in a positive correlation of about 80% to provide more detailed information about short periodic and finer spatial structures. The derived geostrophic velocity components are evaluated by in-situ surface drifter observations. Summarizing all drifter observations in equal-sized bins and comparing the velocity components shows good agreement in spatial patterns, magnitude and flow direction. Mean differences of 0.004 m/s in the zonal and 0.02 m/s in the meridional component are observed. A direct pointwise comparison between the drifter





trajectories and to the drifter location interpolated combined geostrophic velocity components indicates that about 94% of all residuals are smaller than 0.15 m/s.

The dataset is able to provide surface circulation information within the sea ice area and can be used to support a deeper comprehension of ocean currents in the northern Nordic Seas affected by rapid environmental changes in the 1995-2012 time
period. The data is available at https://doi.pangaea.de/10.1594/PANGAEA.900691 (Müller et al. (2019)).

# 1 Introduction

Water masses flowing northwards and southwards through the Greenland Sea and Fram Strait represent the major pathways of the bidirectional water exchange between the Arctic Ocean and the Global Conveyor belt. Most of the water masses are transported via the northwards flowing West Spitsbergen Current (WSC) and the southwards flowing East Greenland Current
(EGC). More than 60% of the total water transport is based on geostrophic movements, caused for example by water density and sea level elevation variations (Rudels, 2012).

Geostrophic currents (GC) can be directly derived from measurements of the dynamic ocean topography (DOT) with respect to the Earth's gravity field, rotation and the Coriolis force involved. In contrast to hydrographic pressure, temperature and salinity observations, collected by irregularly distributed in-situ data (e.g. ARGO floats or ship based measurements), satellite
altimetry is the only possibility to obtain spatially and temporally homogeneous information about the global geostrophic circulation. In-situ sampling platforms can deliver high-resolution measurements, but in polar regions their availability is limited, due to a sparse spatial coverage and challenging environmental conditions. However, especially in sea ice areas, even by altimetry derived geostrophic ocean currents suffer from irregular sampling and data gaps. Furthermore, the generation of a dataset requires some sort of interpolation or gridding techniques, which cause smoothing effects and a coarser spatio-temporal
resolution. Moreover, in open ocean regions, beyond the sea ice edge, the spatial coverage of altimetry data is sparse, due to the along-track acquisition geometry with constant and fixed orbit patterns. Hence, studies are limited to long-term means (e.g. Farrell et al. (2012)) or to satellite altimetry missions dedicated to sea ice conditions (e.g. CryoSat-2; Kwok and Morison (2015) and ICESat; Kwok and Morison (2011)). Nevertheless, monthly DOT estimates have been generated and published by Armitage et al. (2016) using DOT observations derived from long-temporal satellite altimetry. Furthermore, Armitage et al.
(2017) presented a dataset based on a twelve-year altimetry observation (from 2003 to 2014) of geostrophic currents at a monthly time frame on a $0.75° \times 0.25°$ longitude-latitude regular data grid up to a latitudinal limit of 81.5°N. The authors created a dataset, which combines satellite altimetry observations from ice-covered and open ocean regions. Further public available geostrophic ocean current products based on observational data from satellite altimetry only and in combination with in-situ buoys (e.g. Rio et al. (2014)) are provided, for example, by the GlobCurrent project and by the E.U. Copernicus Marine
Service Information (CMEMS). However, the latter's datasets are limited to open ocean conditions.

Besides observation-based ocean circulation products, model simulations provide information of the ocean dynamics. Ocean models differ in spatio-temporal resolutions, forcing model background and underlying mathematical functions. Recent developments are focusing on so-called unstructured ocean models, allowing for locally highly refined spatial resolutions (Danilov





(2013)), while keeping a coarser resolution in other regions of the Earth (e.g. FESOM, Wang et al. (2014) or MPAS-Ocean model, Ringler et al. (2013)). One of the unstructured model is the Finite Element Sea ice Ocean Model (FESOM). For the northern Nordic Seas, an eddy-resolving configuration has been developed, enabling the simulation of small-scale eddies down to 1 km (Wekerle et al. (2017)). Besides total ocean current velocities including wind-driven and geostrophic components, FE-

SOM includes sea surface heights with respect to the bottom ocean topography, which can be also seen as an estimation of the dynamic ocean topography. Applying the gradient of these differential water elevations leads to the computation of simulated geostrophic currents. In contrast to observational based data, models show consistent spatio-temporal resolutions and enable investigations of ocean surface currents under the sea ice layer. However, they are limited to a fixed defined mathematical background and function as an assumption to the reality.

The current publication aims to present an innovative combined data product, based on the advantages of both, simulated and observed datasets. In contrast to other commonly used datasets or assimilation methods, the introduced product is mainly focused on the observational side by filling in modeled DOT elevations, where altimetry data is missing or corrupted. Several investigations and consistency checks have been made by Müller et al. (2019) concluding with a good agreement of simulated and observed DOT in terms of the most dominant seasonal signals and spatial patterns aiming at a combination of the

temporal variability provided by altimetry along-track derived DOT elevations with simulated spatio-temporal homogeneous DOT heights of the model. The combined dataset obtained is characterized by the spatial homogeneous resolution of the model and the temporal variability of altimetry derived DOT elevations. This enables further in space and time consistent studies of geostrophic surface currents in sea ice regions and may help to deepen the knowledge about polar ocean current dynamics.

The dataset is based on a combination of multi-mission satellite altimetry data from the ESA missions Envisat as well as

ERS-2 and the eddy-resolving model, FESOM, covering a period of about 17 years. The combination approach is based on the common known Principal Component Analysis (e.g. Jolliffe (2002), Preisendorfer (1988)), which is successfully applied in historic sea level analyses and reconstruction investigations (e.g. Ray and Douglas (2011), Church et al. (2004)).

The study area covers the Fram Strait region, the Greenland Sea and parts of the Norwegian Sea as well as the Barents Sea. The different regions are summarized by northern Nordic Seas. In geographical coordinates the investigation area is limited to

-30° W to 30° E and 72° N to 82° N.

The paper is structured in four sections. First, the datasets and combination method are introduced, followed by the results. Furthermore, the combination's reliability is evaluated by comparing the obtained datasets with in-situ drifter velocities and independent satellite derived DOT products. The study closes with a summary and concluding remarks of the most significant aspects.

## 2   Data

### 2.1   Observations: Radar altimetry data

The observational part of the combination is provided by high-frequency along-track satellite derived dynamic ocean topography data of the ESA satellites ERS-2 and Envisat. The missions cover a period of about 17 years (1995-05 - 2012-04) up to

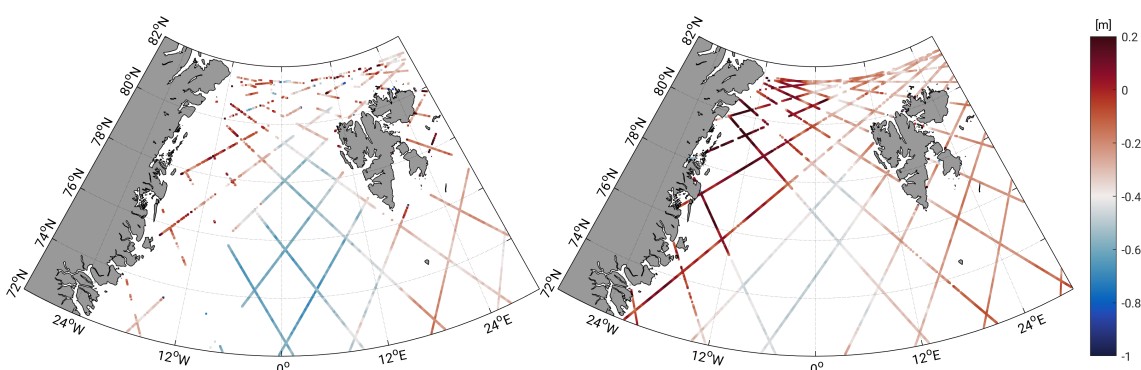

**Figure 1.** Exemplary pre-processed altimetry along-track DOT estimates for Envisat 3-day subcycle in March 2004 (left) and July 2006 (right), illustrating season-dependent data coverage.

a latitudinal limit of 81.5°N. The data pre-processing from ERS-2 and Envisat observed ranges to derived DOT heights and follows the descriptions of Müller et al. (2019). Altimetry ranges are retracked by ALES+ (Passaro et al. (2018)) and open water/sea ice discriminated by applying the method of Müller et al. (2017). The obtained sea surface heights are reduced to DOT estimates by subtracting the high resolved and with in-situ data combined Optimal Geoid Model for Modeling Ocean

Circulation (OGMOC), developed up to a harmonic degree of 2190 (Gruber and Willberg (2018)). ALES+ has been chosen as an optimal retracking algorithm due to the ability for a consistent range estimation independent of the backscattering surface (open-ocean,lead and polynya). Coarse outliers are excluded from the dataset by filtering the sea surface heights on the basis of sea level anomalies (i.e. sea surface heights minus a mean sea surface) before transforming them to physical DOT heights. A time mean inter-mission offset is removed by taking the Envisat time series as a reference within a 6-months overlap pe-

riod (2003/01 - 2003/06) considering only height observations from ice-free regions in the southern part of the investigation area. Before introducing the altimetry DOT elevations to the further processing steps, the ellipsoid referenced observation coordinates are transformed to consider the spherical Earth representation of the model.

Figure 1 shows as an example 3 days of altimetry data during the winter (March 2004) and summer (July 2006) season. In the winter, big data gaps can be noticed close to the East Greenland coast due to the presence of sea ice in contrast to summer,

when most of the data is available.

### 2.2  Simulation: Finite Element Sea ice Ocean Model (FESOM)

The second part of the combination consists of simulated differential water heights (e.g. Figure 2) with respect to the ocean bottom topography (i.e. bathymetry). The bathymetry acts as geopotential surface, which enables a linkage to the altimetry derived DOT heights (Androsov et al. (2018)). FESOM is a global multiresolution circulation model with an included sea ice

component resolving the major sea ice drift patterns. The model is based on the standard set of hydrostatic primitive equations in the Boussinesq approximation (Wekerle et al. (2017)) and is characterized by an unstructured triangular mesh with 47 vertical levels (Wang et al. (2014)). The horizontal resolution in the configuration used in this study reaches up to 1 km in

(c) Author(s) 2019. CC BY 4.0 License.



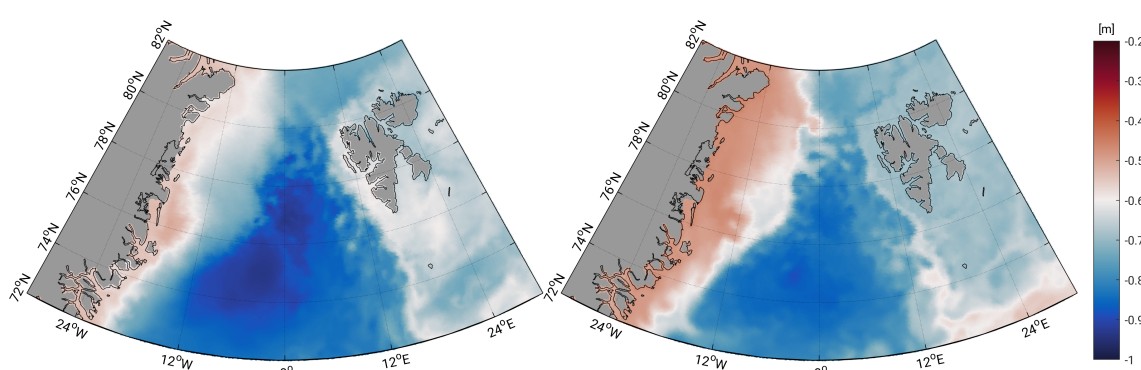

**Figure 2.** Exemplary differential water heights in March 2004 (left) and July 2006 (right) simulated by FESOM. Note the offset in comparison to Figure 1.

the Fram Strait and northern Greenland Sea area and can be described as "eddy-resolving". Furthermore, the geographical model coordinates are referenced to a spherical Earth representation with a radius of 6,367.50 km. More details of the FESOM configuration can be found in Wekerle et al. (2017). The present study uses only daily differential water heights (DWH) of the surface level covering the period 2002-2009.

**2.3 Comparative datasets**

For validation a comparison with external generated absolute dynamic topography (ADT) elevations, from ADT derived geostrophic velocity components and to geostrophic ocean velocities reduced in-situ drifter observations is performed. The ADT data including geostrophic velocity components (Pujol and Mertz (2019)), provided by the E.U. Copernicus Marine Service Information platform (CMEMS) are characterized by a daily and 1/4 degree spatial resolution and are based on multi-
mission altimetry data. The ADT grids are created by adding temporal variable sea level anomalies to a mean dynamic topography and cover the complete time period of the developed datasets. However, no ADT and current data are available in sea ice areas, which limits the comparison to ice-free regions and seasons.

Further interpolated surface drifter trajectories from CMEMS (Rio and Etienne (2018)) with a 6-hour interval are used. Following the pre-processing steps of the drifting buoys, described by Rio and Etienne (2018), all surface drifters are analyzed
concerning their drogue status and local wind slippage corrections. Besides geostrophic velocities, drifter observations include a-geostrophic movements (e.g. Ekman currents, Stokes drift, inertial oscillations, local wind effects, etc.). Hence, the drifter data must be corrected enabling a comparison with satellite altimetry and simulated derived geostrophic currents. Local wind corrections, also provided by CMEMS (Rio and Etienne (2018)), are directly subtracted from the drifter velocities, considering the drogue status. The Ekman current is taken from global grids providing velocities at 15m depth (drogue on) and at the surface
(drogue off) level. The computation of the Ekman fields follows the explanations and processing scheme of Rio and Hernandez (2003) and Rio et al. (2014). The 3-hourly available Ekman grids are downloaded from the GlobCurrent data repository and have a spatial resolution of 1/4 degrees and a global coverage. However, grid nodes north of 78.875°N are not defined, which





limits the comparison to central parts of the Greenland Sea and neglects the Fram Strait area. The Ekman velocities are interpolated to the drifter positions and subtracted from the drifting buoys velocity by taking the drogue status into account. The Stokes drift is provided globally (Rascle and Ardhuin (2013) distributed by GlobCurrent) and applied only to undrogued surface drifter data in the same way like the Ekman fields (Rio et al., 2014). Following the suggestions of Andersson et al.

(2011), the Ekman and Stokes drift reduced drifter velocities are low-pass filtered by a 25-hour cutoff, two-point Butterworth filter to remove tidal and inertial oscillations. Furthermore, drifters showing observations with time gaps of more than 1 day are filtered separately (Andersson et al. (2011)).

Most of the drifter buoys observations are collected in ice-free regions affected by currents (see Figure A1). Analyzing the geostrophic amplitudes and phases, the major pathway and stream velocity of the West Spitsbergen Current is clearly

identified, in contrast to the East Greenland Current, which is mostly covered by sea ice. Due to high variability, most of the drifter data can be found in the West Spitsbergen region and in the southern parts, where Atlantic water enters the Greenland Sea. Most of the drifting buoys are carried through the Fram Strait or enter the Barents Sea. Only a few drifter buoys turn around and follow the East Greenland Current. Furthermore, smaller eddies in the central Greenland Sea can be observed. In this study, nearly 70,000 in-situ observations are available, of which 63% are characterized by a drogue on status. The number

of drifter measurements strongly increases between 2007 and 2012. However, nearly no data can be used between 2000 and 2006. Nevertheless, a validation of the ERS-2 data products is possible between 1995 and 2000.

## 3   Method

In order to generate a combined spatio-temporal consistent dataset based on irregular distributed altimetry observations, it is necessary to connect the along-track derived DOT estimates with a spatially consistent modeled DOT representation to fill the

observation gaps. The following chapter describes briefly the combination of along-track DOT heights with the modeled water level, while keeping the spatial height reference of the altimetry observations.

The combination is mainly based on a Principal Component Analysis (PCA) transferring the method of historic sea level reconstruction (e.g. Church et al. (2004), Ray and Douglas (2011)) to the present purpose. Altimetry observed along-track DOT heights represent the temporal DOT variability, whereas the spatial signal is provided by FESOM. Figure 3 highlights the

interrelationship of the datasets and gives an overview over the main processing chain. The individual work steps are described chronologically. The output of the processing steps are combined geostrophic currents (cGC) and dynamic ocean topography (cDOT) data representing the temporal variability of the altimetry measurements and the spatial homogeneity of the ocean model.

### 3.1   Data pre-processing

The input of the data production chain are along-track DOT elevations and daily simulated finite element formulated differential water heights (DWH). In order to establish an equal combination basis, both datasets are treated equally. First, they are reduced by their time mean offsets and the most dominant seasonal (i.e. annual) signal (Müller et al. (2019)).

Open Access · Earth System **Science** **Data** Discussions

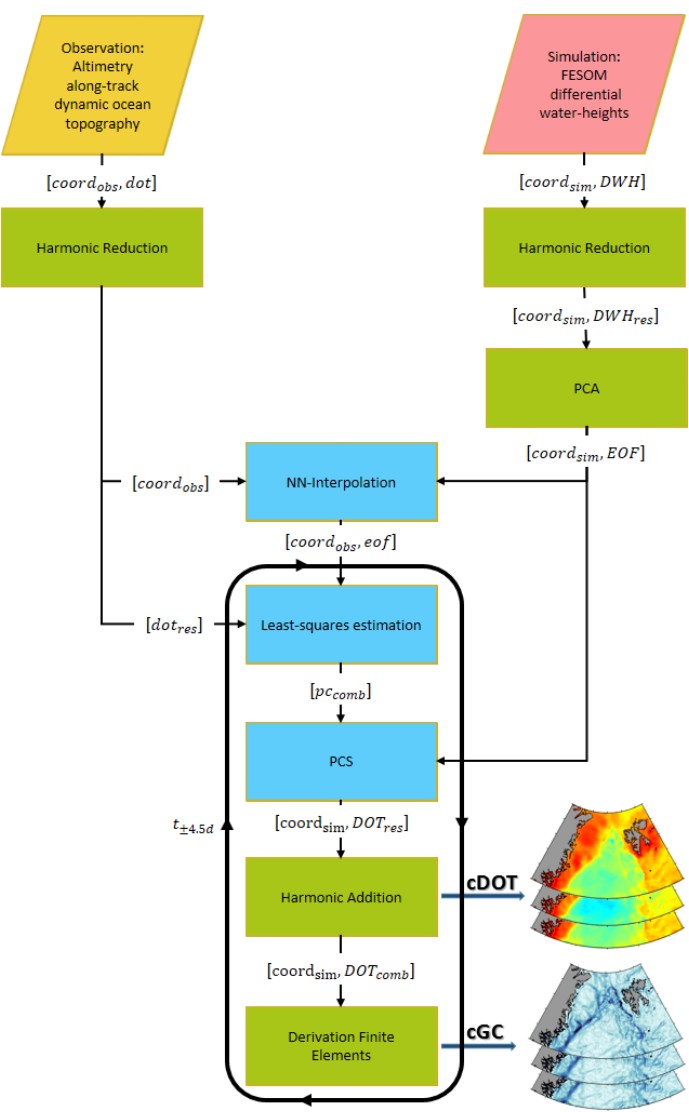

**Figure 3.** Flow chart of combined dynamic ocean topography (cDOT) and geostrophic currents (cGC) product, based on harmonic reduction, nearest-neighbor interpolation (NN-Interpolation), principal component analysis (PCA) and synthesis (PCS). Light blue indicates the combination and green auxiliary processing steps. The data sources are highlighted in yellow and pink. Data labeled in capitals indicate a grid and small letters a representation on ground tracks. Indices describe simulated ($sim$), observed ($obs$), combined ($comb$) and by annual signal and constant offset reduced ($res$) datasets. $t_{\pm 4.5d}$ represents the 9-day time span of used altimetry observations.

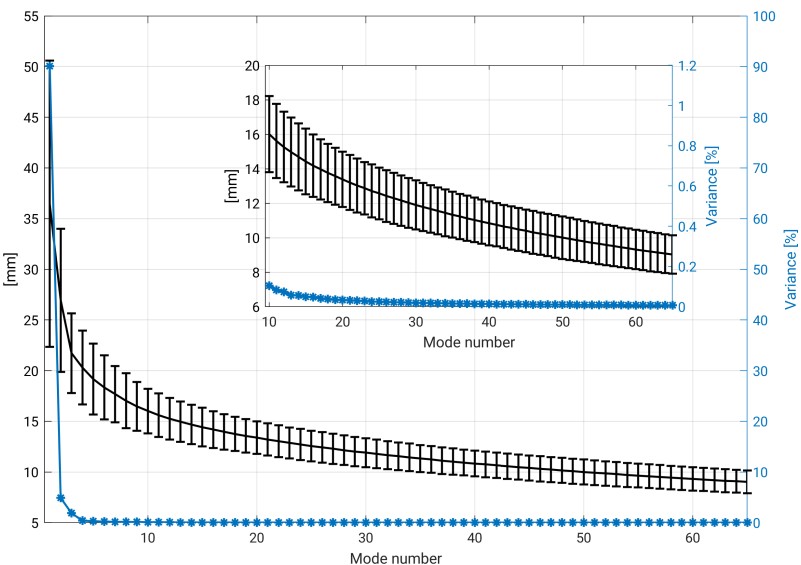

**Figure 4.** Percentage of variance (blue) and daily averaged root mean square error (black) including standard deviation of FESOM original data and reconstructed signal for 65 Modes of Principal Component Synthesis. For better overview, Modes 10-65 are zoomed in.

In a second step, the reduced FESOM grids are introduced to a PCA in order to decompose them in a linearly uncorrelated, temporal part (i.e principal components) describing the temporal evolution, and in empirical orthogonal functions (EOF) identifying most dominant spatial structures of the time series. They are sorted in a decreasing order with respect to their contribution to the total signal variance. In order to reconstruct the original signal, the principal components and the corresponding EOFs

have to be multiplied and summed up. The product of one combination pair is called Mode. This inverse process of PCA is also called Principal Component Synthesis (PCS). Not necessarily PCS is used to reconstruct always the full signal, however the approach can be also limited to a certain number of retaining Modes, representing a significant percentage of the total signal. Mathematical and functional relations are explained in Jolliffe (2002). In order to determine the number of the most significant EOFs, the root mean square error (RMSE) is computed for comparing the original FESOM DWH and the reconstructed signal.

The RMSE is computed by (Barnston (1992)):

$$RMSE(t) = \sqrt{\overline{(l_t - r_t)^2}} \tag{1}$$

where $l$ substitutes the original FESOM DWH and $r$ the reconstructed grids of the day $t$, where the overbar is computed over all grid nodes. Figure 4 shows the evolution of the temporal amount of variance and the temporal averaged RMSE with respect to the individual number of Modes. It is decided to use 50 Modes resulting in an RMSE of about 10 mm and a summed variance

of more than 99%. In the following processing steps, only the spatial signals (i.e. EOFs) of FESOM are used. In contrast, the principal components, describing the temporal evolution of the different modes, are neglected.





## 3.2 Combination

The combination step links the pre-processed along-track DOT heights with the most significant spatial pattern obtained from the PCA of the FESOM differential water heights. The processing is based on a daily temporal resolution, including 9 days of radar altimetry data for each time step. The time steps are referred to the mean of a 9-day time span (i.e. $t_{\pm 4.5d}$). The combined DOT heights (cDOT) can be represented by a linear combination of $n$ combined estimated principal components and the obtained EOF grids from FESOM. The functional relation of the PCS is described in Equation 2:

$$cDOT_{res}(x,y,t) = \sum_{i=1}^{n} pc_{comb_i}(t) \cdot EOF_i(x,y) \qquad (2)$$

where $n$ corresponds to the number of significant principal components and empirical orthogonal functions. $pc_{comb_i}$ substitutes the $n$ unknown combined principal components and $EOF_i(x,y)$ the $n$ most dominant spatial pattern on the FESOM grid (see sec. 3.1).

The principal components ($pc_{comb_i}$) are estimated by fitting the model EOFs to the altimetry derived DOT elevations $dot_{res}$. Therefore, the EOF grids are interpolated to the observation coordinates based on NN-Interpolation resulting in along-track sampled empirical orthogonal functions ($eof_i(x,y)$). The solution for $pc_{comb_i}$ is then given by applying the least squares method (e.g. Koch (1999)) to Equation 3:

$$dot_{res}(x,y,t_{\pm 4.5d}) = \sum_{i=1}^{n} pc_{comb_i}(t) \cdot eof_i(x,y) \qquad (3)$$

where $dot_{res}(x,y,t_{\pm 4.5d})$ includes all altimetry derived DOT heights within $\pm 4.5$ days and $eof_i(x,y)$ the corresponding along-track interpolated modeled EOFs. The result are $n$ time series of combined principal components. Related detailed background information to solve for $pc_{comb_i}$ is prepared in Appendix A.

Furthermore, a Gaussian weighting, which considers uncertainties in the altimetry DOT heights due to the presence of sea ice, is introduced to the least squares process. The individual weights are defined by using external sea ice concentration from the National Snow and Ice Data Center (NSIDC, Fetterer et al. (2017)) interpolated to observation coordinates and take an enhanced error budget of altimetry range estimations, due to noisier observations taken into account within the sea ice area. In a last step, the estimated principal components are introduced to the PCS (Equation 2) in order to construct a combined DOT solution ($cDOT_{res}(x,y,t)$). The individual combination steps are outlined in Figure 5 and are briefly summarized in chronological order as follows:

1. Separation of reduced FESOM DWH into most dominant spatial patterns ($EOF$) and time series of principal components applying PCA. However, the principal components obtained are not used but neglected, since new principal components are estimated from altimetry derived DOT and most dominant spatial patterns (EOF) of FESOM in the further combination steps.


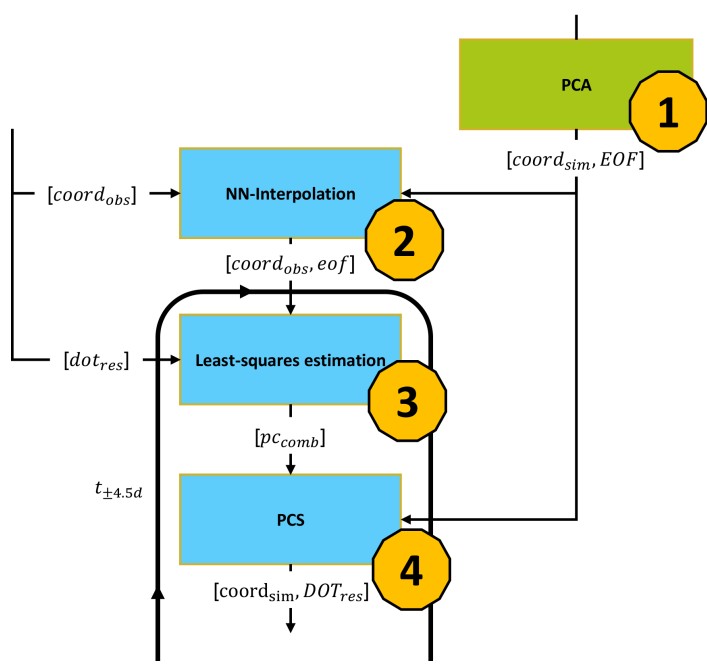

**Figure 5.** Subset of Figure 3 outlining combination steps. Numbers indicate the chronological order of the individual processing steps.

2. Nearest-neighbor interpolation of $EOF$ to altimetry along-track observations ($coord_{obs}$) obtaining profiled $eof$.

3. Least squares estimation (see Appendix A) of combined principal components ($pc_{comb}$) by solving Equation 3 based on altimetry DOT observations ($dot_{res}$) and interpolated empirical orthogonal functions ($eof$).

4. Application of Equation 2 to obtain the combined DOT ($cDOT_{res}$) dataset in the FESOM grid ($coord_{sim}$) based on
5     $pc_{comb}$ (step 3) and $EOF$ (step 1).

### 3.3    Data generation

In order to reconstruct the full signal and to rescale the combined heights to the altimetry height reference, the previous subtracted altimetry time mean offset and annual signal are re-added (Sect. 3.1). In the next step, combined geostrophic currents (cGC) are obtained by computing the zonal ($u_g$) and meridional ($v_g$) geostrophic velocity components at the surface, given by
10    Equation 4:

$$u_g = -\frac{g}{f}\frac{\partial h}{\partial y}$$
$$v_g = \frac{g}{f}\frac{\partial h}{\partial x}$$
(4)





where $g$ is the acceleration of gravity ($9.832 \frac{m}{s^2}$), $f = 2\Omega \sin\phi$ the Coriolis force, $\phi$ the latitude and $\Omega$ the Earth's rotation rate. $\partial h$ denotes the horizontal gradient in $x$ and $y$ direction of cDOT height $h$. The derivatives $\frac{\partial h}{\partial y}$ and $\frac{\partial h}{\partial x}$ are solved based on the finite element method (see Appendix B) which prevents further smoothing effects. Furthermore, the geostrophic absolute velocity ($A_g$), phase $\phi_g$ and eddy kinetic energy (EKE) can be computed by applying Equation 5.

$$
\begin{aligned}
A_g &= \sqrt{u_g^2 + v_g^2} \quad \phi_g = \arctan \frac{v_g}{u_g} \\
EKE &= \frac{1}{2}((u_g(t) - \overline{u_g})^2 + (v_g(t) - \overline{v_g})^2)
\end{aligned}
\tag{5}
$$

where $t$ substitutes the velocity at a certain time and the overbar indicates the mean velocity for a defined time period (e.g. quarterly).

## 4 Datasets

The combined DOT and geostrophic current velocity fields contain DOT heights derived from satellite altimetry and simulated differential water heights from FESOM (Müller et al. (2019)). The dataset spans a time period from mid-May 1995 to early April 2012 and cover the investigation area of the northern Nordic Seas limited to -30° W to 30° E and 72° N to 82° N. The datasets are saved in NetCDF format. As a result of the combination process, the processed grids are stored in a daily temporal and unstructured spatial resolution with local refinements up to 1 km. Missing days in the dataset due to longer periods of missing altimetry observations are possible. Furthermore, an outlier detection based on the accuracy of the computed combined principal components is performed rejecting erroneous combination estimations. The data product is given in units of meters in case of DOT and in m/s for the geostrophic components.

Figure 6 illustrates quarterly averaged daily combined DOT heights and derived geostrophic components expressed in velocity and azimuth. All meshes show the same spatial resolution with local refinements in the central Greenland Sea and Fram Strait region (approx. 1 km) and implicate the Finite Element structure of the input model. The three-monthly averaged cDOT fields vary by circa 1 m across the northern Nordic Seas with maximum variations in the winter months. Furthermore, the anti-phase relationship in the annual oscillation (Bulczak et al. (2015)) between the deep basins and the shelf areas in winter and summer can be seen. The derived geostrophic components show a strong meandering West Spitsbergen Current and a more clear flow structure in the East Greenland Current.

## 5 Comparison with external datasets

The produced datasets are compared with independent datasets providing daily sampled DOT heights and observations of surface drifter buoys. However, it must be noted that the comparison is challenging since no dataset can be used as ground truth in the whole area of study.

In order to follow a comparison with in-situ observations, the combined geostrophic components are spatio-temporally interpolated to surface drifter locations. This enables the analyses of differences between geostrophic currents from observations

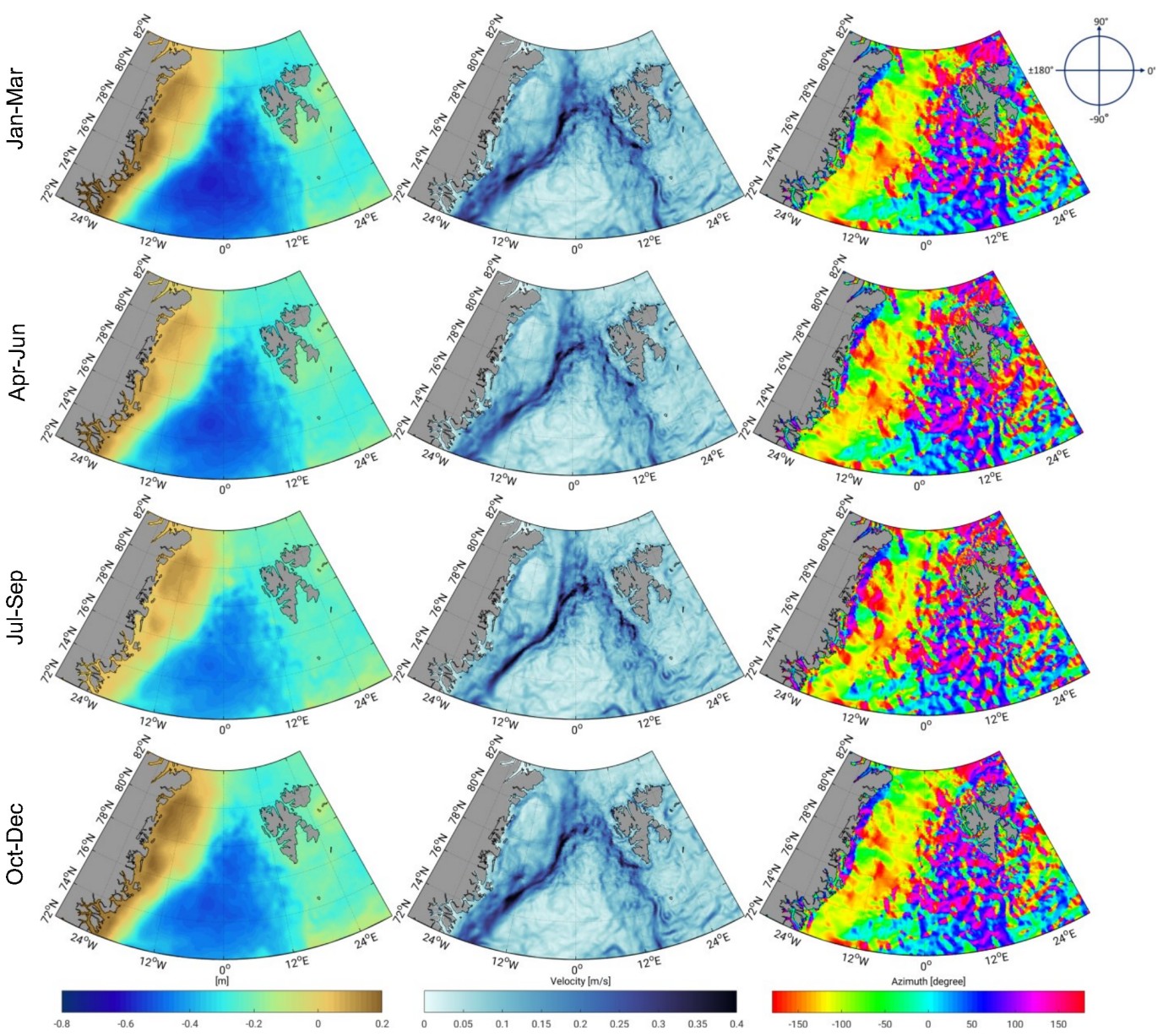

**Figure 6.** Three-monthly averaged combined DOT heights (left), absolute geostrophic velocities (middle) and flow direction (right) from 1995 to 2012.





and from the derived combined product. Therefore, the combination procedure is applied to the drifter epochs. This is done by interpolating the estimated combined principal components linearly to the drifter times followed by a PCS (eq. 2) and a spatial nearest-neighbor interpolation to the drifter location. The result are combined DOT heights at the drifter observation time and location. In order to compare with the geostrophic drifter measurements, the cDOT heights are transformed to geostrophic

velocities (sec. 3.3).

Following Andersson et al. (2011), the drifter observations are grouped into 2°x 1° longitude latitude boxes. In order to per- form statistically reliable analyses, only bins with at least 2 different surface drifters and 50 observations are used (Andersson et al. (2011)).

Figure 7 displays in time-averaged u and v components of the drifter observations (first row) and the combined geostrophic

currents (middle row). The differences (bottom row) agree well with spatial patterns of the velocity components (i.e. drifter - combination). The East Greenland and West Spitsbergen Current are resolved by both datasets in both velocity components. The drifter and the cGC describe the same amplitude and flow direction in most of the bins. However, the v component shows bigger differences than the zonal component, caused mainly by a higher variability due to the primarily meridional flow direction of the currents in this area. Good agreement to the drifter data is shown by slight mean differences of 0.004 m/s

±0.02 m/s in the zonal (u) and 0.01 m/s±0.04 m/s in the meridional (v) component.

Computing directly the RMSE based on the individual trajectories between the drifter and combined velocity components for each drifter, a mean of 0.127 m/s±0.034 m/s in case of the u-velocity and 0.132 m/s±0.039 m/s for the v-velocity are obtained. Moreover, the RMSE may reach 0.225 m/s for u and 0.232 m/s for v. Higher RMSE values can be found in regions with strong current activity (e.g. WSC).

Figure 8 shows the RMSE distribution of absolute velocity (Equation 5) for the period 1995-2012 (blue curve). In addition, the same quantity derived based on the altimetry-only ADT currents is plotted in green. Both datasets are characterized by a very similar behavior. Nevertheless, the combination shows smaller residuals. 35% of the combined residuals are smaller than 0.1 m/s in contrast to 27% of the altimetry only derived geostrophic absolute velocity. In general, the results of both datasets are comparable to previous studies of the World Ocean and to Volkov and Pujol (2012) describing a maximal RMSE of around

0.2 m/s and a typical range of 0.07 m/s to 0.15 m/s for the northern Nordic Seas in both components.

Figure 9 shows daily three-monthly averaged EKE of the combined and ADT grids within the investigation period (1995- 2012). The EKE results are computed by subtracting three-monthly means from the daily datasets (Equation 5). The ADT appears smoother and shows big data gaps in sea ice regions in comparison with the combined results. Furthermore, the com- bined eddy fields show finer eddy structures within the sea ice area and close to the Greenland coast. The cGC are characterized

by a higher spatial resolution and more variability in current regions.

The cDOT grids are evaluated against the daily and spatial averaged time series of ADT fields. Therefore, the cDOT fields are spatially interpolated to the ADT grids. Figure 10 shows the by their mean reduced temporal evolution of both datasets. The comparison covers the full investigation period, but spatially limited to ice-free regions. The time series indicate a positive temporal correlation of nearly 80%. Both datasets display high-frequent patterns. Compared to the stronger smoothed ADT

grids with a standard deviation (std) of ±0.04 m, the cDOT heights are characterized by a higher variability (std= ±0.05 m)

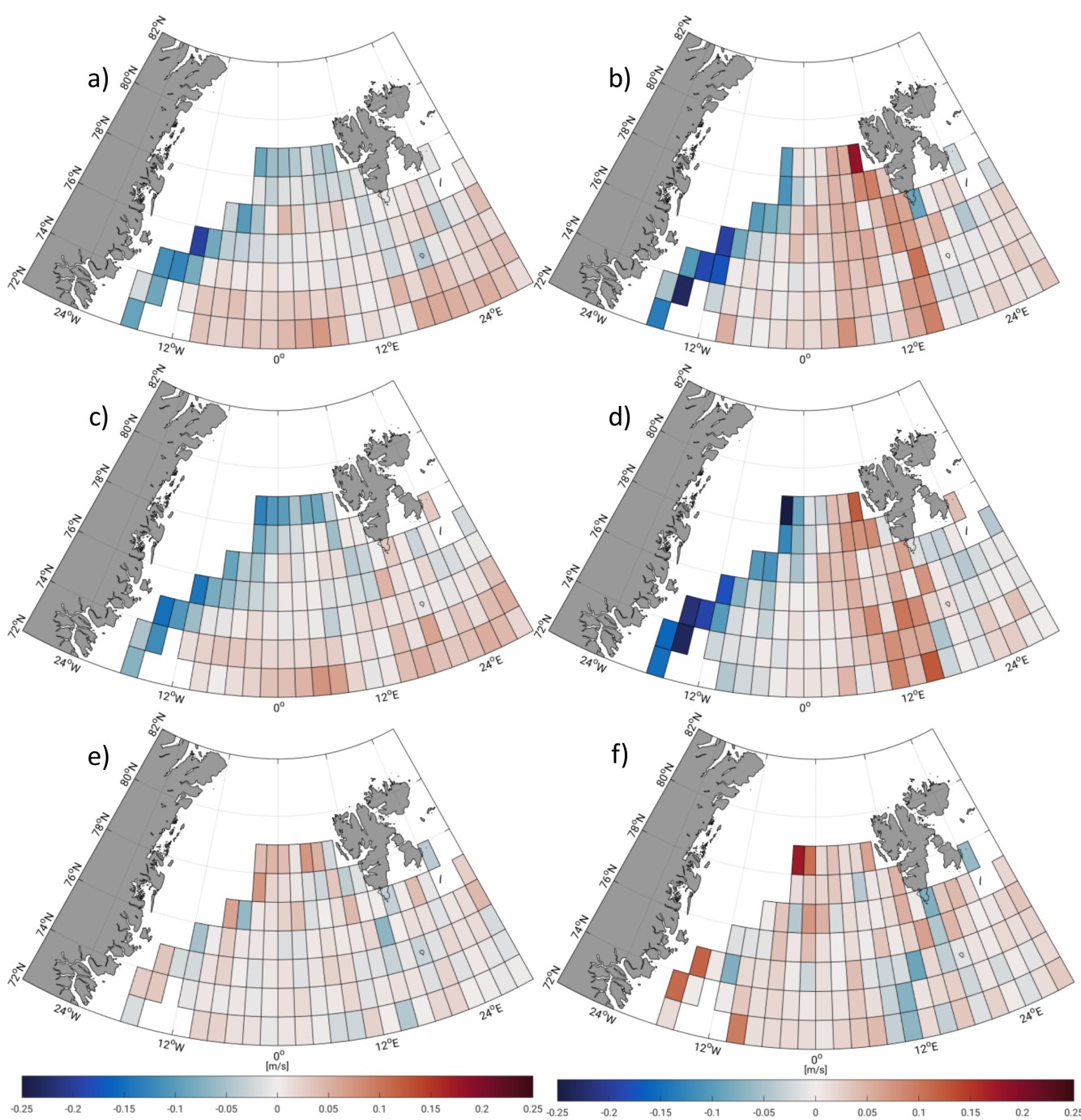

**Figure 7.** Temporal averaged geostrophic u (left) and v (right) components of drifter observations (a,b), combined dataset (c,d) and differences (e,f), respectively, binned in 2°x1°(lon,lat) boxes within the investigation time (1995-2012). Please note the changed scale of the differences.



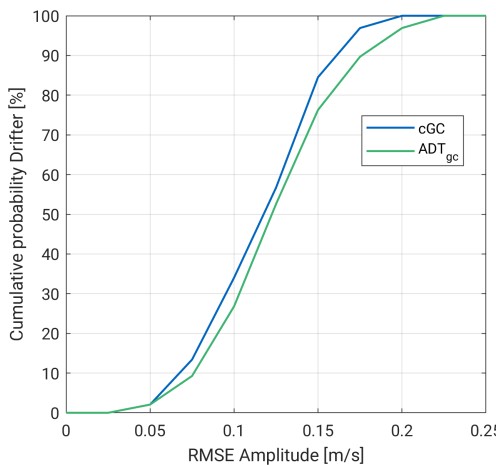

**Figure 8.** RMSE of geostrophic absolute velocity between drifter observations and to the trajectories interpolated combined and ADT datasets from 1995 to 2012.

and display short periodic structures. Nevertheless, a slight offset between the time series between 1995 and 2003 of 2.5 cm and 2.0 cm between 2003 and 2012 can be observed, which might occur due to a different applied mean epoch of the ADT computation or an unconsidered bias in the retracking procedure of ERS-2 and Envisat.

## 6 Summary and Conclusions

The current paper presents an innovative dataset based on a combination of height observations from satellite altimetry with spatial information provided by an ocean model (FESOM). In case of altimetry data, an open water classification procedure is applied in order to exploit along-track water height measurements within the sea ice area. Furthermore, height offsets between the open ocean and the sea ice area are removed by using one single retracking algorithm.

The combination approach takes advantage of the principal component analysis, especially the separation of the model data
into its most significant spatial patterns and temporal components with respect of the total variability. The 50 most dominant patterns (EOF) are used to combine them with ERS-2 and Envisat observed along-track DOT heights in order to fill in observational gaps and to enable investigations based on a homogeneous DOT representation. In detail, the spatial information from FESOM and the temporal variability from altimetry are linked. The height level of the final product is given by altimetry by re-adding the previous estimated and subtracted annual signal and constant offset, since the model height reference is not clearly
defined. Whereas the obtained spatial resolution is given by FESOM, characterized by local refinements in ocean current active areas smaller than 1 km. The combination is computed on a daily resolution and covers a time span of 17 years (1995-2012).

Geostrophic currents are provided by computing zonal and meridional slope gradients of the Finite Element mesh. This allows comprehensive variability analyses of ocean currents not only in open ocean areas, but also within sea ice regions. A comparison with altimetry-only datasets shows that the combination convinces by an enhanced spatio-temporal resolution,

**Figure 9.** Three-monthly averaged geostrophic eddy kinetic energy within the FESOM period (1995-2012) for combined results (left) and ADT grids (right). Green areas indicate missing values.



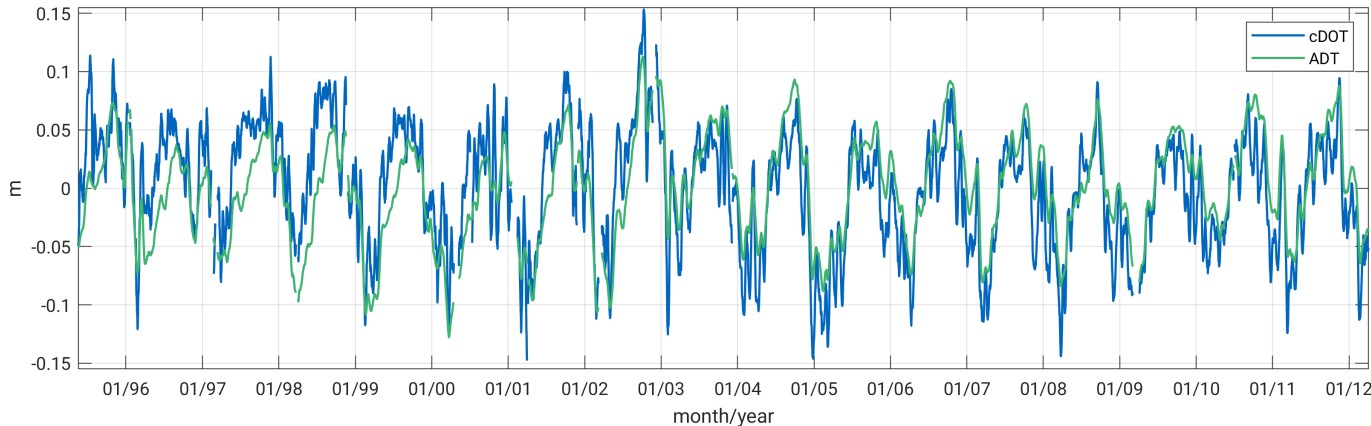

**Figure 10.** Zero-centered time series of daily and spatial averaged altimetry-only ADT grids and to the ADT grid nodes interpolated combined DOT (cDOT) limited to ice-free regions within 1995-2012 and the northern Nordic Seas.

displays short periodic structures and missing data gaps, especially in the regions covered by sea ice. Moreover, a positive correlation of nearly 80% in open ocean areas can be achieved.

A comparison with in-situ surface drifter measurements, although limited to ice-free regions, indicates a similar and realistic representation of ocean current patterns and meso-scale eddies in the area of both datasets under investigation. Furthermore, a

good agreement in the comparison of binned surface drifter and derived combined geostrophic velocity components has been described. A comparison with the uncompressed FESOM geostrophic currents in terms of EKE displays a small improvement by the combined dataset. A direct pointwise comparison for each drifter trajectory indicates a temporal RMS of the differences between the drifter velocity components and the combination of about 0.13 m/s. In general, the RMSE values obtained range from 0.05 m/s to 0.10 m/s in areas with low-flow activity and up to 0.22 m/s in regions with high current energy. Following

Volkov and Pujol (2012), this velocities are comparable to previous estimates for the World Ocean.

The presented data product supports long-temporal studies of the dynamic ocean topography and the ocean current regime in polar regions affected by sea ice. Aiming at a more than 25 years covering extension of the dataset more conventional altimetry (Saral, ERS-1) as well as Delay-Doppler altimetry data (e.g. Sentinel-3A/B, CryoSat-2) will be added to the combination process.

**7   Data availability**

The final combined dataset can be downloaded from PANGAEA, https://doi.pangaea.de/10.1594/PANGAEA.900691 (Müller et al. (2019)). Envisat (SGDR) and ERS-2 (REAPER-SGDR) altimetry data access is available from ESA (Envisat, https://doi.org/10.5270/EN1-85m0a7b, ESA (2018); ERS-2, https://earth.esa.int/web/guest/news/-/article/reprocessed-esa-ers-altimetry-reaper-dataset-now-available, Brockley et al. (2017)). FESOM data can be downloaded

from PANGAEA, https://doi.org/10.1594/PANGAEA.880569 or requested from AWI. The in-situ drifter observations



and ADT grids with additional parameters are available via CMEMS (drifter data, http://resources.marine.copernicus.eu/
?option=com_csw&view=details&product_id=INSITU_GLO_UV_L2_REP_OBSERVATIONS_013_044, Rio and Etienne
(2018); ADT, http://resources.marine.copernicus.eu/?option=com_csw&view=details&product_id=SEALEVEL_GLO_PHY_
L4_REP_OBSERVATIONS_008_047, Pujol and Mertz (2019)). Data grids of the Ekman- and Stokes drift are provided by
GlobCurrent http://globcurrent.ifremer.fr/products-data/data-catalogue.

*Acknowledgements.*   The authors thank Hélène Etienne from Collecte Localisation Satellites (CLS) for supporting the study with valuable
advice regarding the surface drifter data processing. The authors thank the Chair of Astronomical and Physical Geodesy, Technical University
of Munich (TUM) for providing the geoid model, OGMOC. We also thank ESA for providing ERS-2 and Envisat altimetry data, CMEMS
for maintaining the ADT grids as well as the surface drifter velocities and GlobCurrent for providing a-geostrophic currents (i.e. Ekman,
Stokes drift). This work was mainly supported by the German Research Foundation (DFG) through grants, BO1228/13-1 and DE2174/3-1.
The publication is funded by the German Research Foundation (DFG) and the Technical University of Munich (TUM) in the framework of
the Open Access Publishing Program.

*Author contributions.*   Felix L. Müller processed the combination and wrote most of the paper. Denise Dettmering supervised the present
study, contributed to the manuscript writing and helped with discussions of the results. Claudia Wekerle provided the FESOM data and
contributed to the manuscript writing. Christian Schwatke maintains the altimetry data base (OpenADB) at DGFI-TUM and supports with
discussions. Marcello Passaro developed the retracking algorithm and helped with discussion concerning the altimetry dataset. Wolfgang
Bosch initiated the study. Florian Seitz supervised the research.

*Competing interests.*   The authors declare no conflict of interests.

## Appendix A: Estimation of combined principal components

The solution for $pc_{comb_i}$ is given by applying least squares (Koch (1999)):

$$l + v = E\beta$$
$$\beta = (E^T W E)^{-1} \cdot E^T W l \tag{A1}$$

where $\beta$ substitutes $pc_{comb_i}$, $l$ the observations $dot_{res}$ and $E$ the interpolated empirical orthogonal functions, $eof$. The
estimated residuals are represented by $v$. $W$ indicates a diagonal weighting matrix. $T$ denotes transposed matrices.



## Appendix B: Derivation of Finite-Elements in FESOM

The FESOM configuration that was used is based on a finite element formulation. Regarding the spatial discretization, the global ocean is discretized by using tetrahedral elements. These elements are constructed by first generating a surface triangular mesh. In the vertical, z-layers are used. The resulting vertical prisms are then cut into three tetrahedrals. In the finite element method, variables are approximated as linear combinations of a finite set of basis functions $\{N_i\}$. Regarding the choice of these basis functions, FESOM uses a P1-P1 discretization, meaning that piecewise linear basis functions are employed for both sea surface height $\eta$ and horizontal velocity $\mathbf{u}$:

$$\eta = \sum_{i=1}^{N_{2D}} \eta_i N_i \text{ and } \mathbf{u} = \sum_{i=1}^{N_{3D}} u_i N_i,$$

where $N_{2D}$ and $N_{3D}$ denote the number of 2D and 3D nodes respectively. The $i$th basis function $N_i$ is equal to 1 at node $i$ and linearly vanishes to 0 within elements containing this node.

Derivatives are computed by transformation to a reference element. In 2D, we consider the reference element $\hat{K}$ defined by nodes $\hat{a}_1 = (0,0)$, $\hat{a}_2 = (1,0)$ and $\hat{a}_3 = (0,1)$. As local 2D basis functions defined on $\hat{K}$, we choose the first order polynomials

$$N_1(x,y) = 1 - x - y, \ N_2(x,y) = x \text{ and } N_3(x,y) = y,$$

with its Jacobian matrix $J_N = \begin{pmatrix} -1 & -1 \\ 1 & 0 \\ 0 & 1 \end{pmatrix}$. Any arbitrary element $K$ in the physical domain defined by nodes $a_1$, $a_2$ and $a_3$ can be mapped on the reference element $\hat{K}$ by affine-linear transformation:

$$F: \ \hat{K} \to K, \ F(\hat{x}) = B\hat{x} + d,$$

with $B = (a_2 - a_1, a_3 - a_1)$ and $d = a_1$. When computing the gradient of a variable $\phi$ on the reference element $\hat{K}$, we obtain:

$$\nabla_{\hat{x}}\phi(x) = \nabla_{\hat{x}}\phi(F(\hat{x})) = \nabla_x\phi(F(\hat{x}))\nabla_{\hat{x}}F(\hat{x}) = \nabla_x\phi(F(\hat{x}))B,$$

Thus, the gradient in the physical domain can be expressed as:

$$\nabla_x\phi(F(\hat{x})) = \nabla_{\hat{x}}\phi(F(\hat{x}))B^{-1}.$$

We now compute the gradient of $\eta$ on element $K$ by inserting $\phi = \sum_{i=1}^3 \eta_i N_i$ in the above equation:

$$\nabla_x\eta = \nabla_{\hat{x}}\sum_{i=1}^3 \eta_i N_i B^{-1} = (\eta_1, \eta_2, \eta_3)J_N B^{-1}.$$

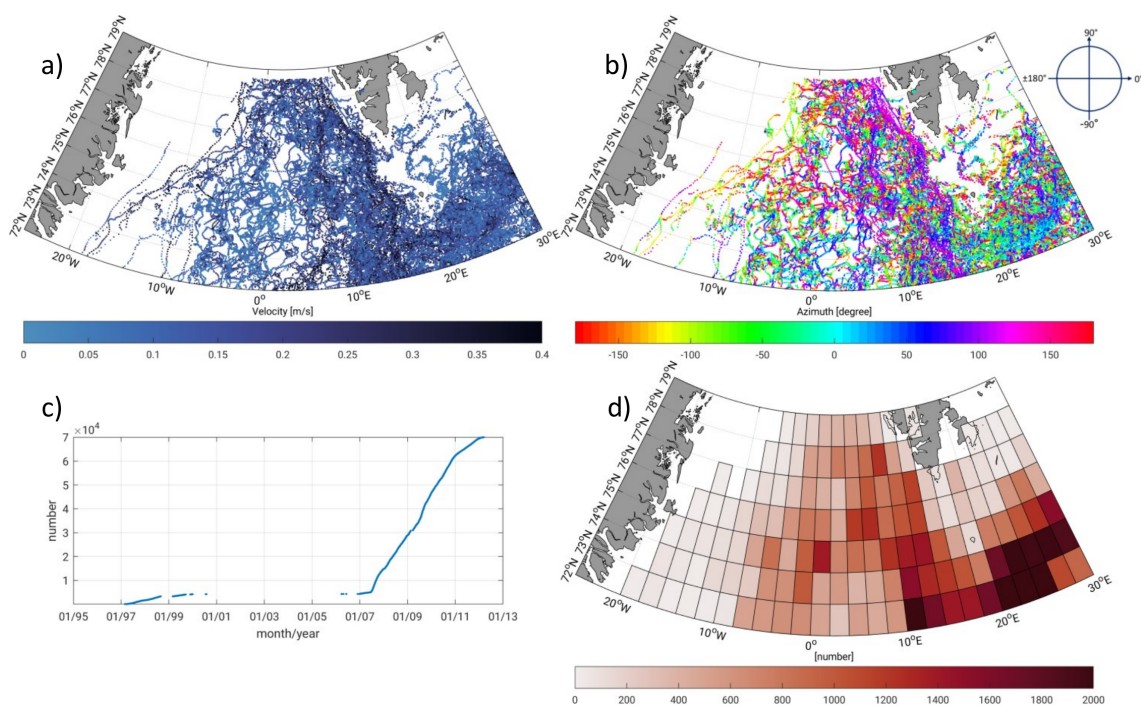

**Figure A1.** Amplitude (a), azimuth (b), cumulative number (c) of geostrophic surface drifter velocities and number of records in 2°x1° boxes (d) within the 1995-2012 investigation time period. Approximately 63% of the observations were obtained by an attached drogue.

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
