# Peer review of "Geostrophic Currents in the northern Nordic Seas from a Combination of Multi-Mission Satellite Altimetry and Ocean Modeling"

_Earth System Science Data, 2019_

## Referee Comment (RC1) · Anonymous Referee #1 · 10 Sep 2019

**1   General comments**

The article describe a new dataset constructed using altimetry, along-track measurements with the outputs a high-resolution numerical model in the "northern Nordic Seas". Overall the article is clearly written and all the details concerning the processing is provided. Nevertheless I formulate 3 main comments regarding the present version of the manuscript:

1. The innovation: the manuscript would be enriched if the authors add a brief justification of the advantages of their new dataset with respect to results of numerical

simulations in the same area.

2. The dataset itself: while it is clear that the FESOM model is based on finite elements, having the different variables on such an unstructured grid may not be the easiest option for most of the users.

3. The notations: starting from the schema (Figure 3), the way the different variables are defined make the developments difficult to follow.

Suggestions are provided in the specific comments. I believe that the overall quality and the readability would be improved if the authors can address these 3 points.

**2 Specific comments**

P1 L9: it is stated that the presented method differs from data assimilation because it substitutes altimetry data with the model output; however it is not clear in the manuscript what is the advantage over the dataset obtained with a pure assimilation approach.

P1 L14: the by altimetry obtained annual signal
→ the annual signal obtained by altimetry

P1 L17: and the temporal variability of the altimetry along-track derived DOT heights
→ specify the temporal resolution (1 day according to the netCDF)

P1 L21: what justifies the difference in the order of magnitude (0.004 vs 0.02 m/s) for the zonal and meridional components?

P2 L1: drifter location interpolated combined geostrophic velocity components
→ combined geostrophic velocity components interpolated onto the drifter locations

P2 L2: a general map with the main geographical features would help the readers unfamiliar with the region of interest.

P2 L18: by altimetry derived geostrophic ocean currents
→ geostrophic ocean currents derived by altimetry or altimetry-derived geostrophic ocean currents

P2 L29: CMEMS stands for "Copernicus Marine Environment Monitoring Service"

P2 L32: and underlying mathematical functions
→ mathematical formulations

P2 L31: information of the ocean dynamics
→ information on/about the ocean dynamics

P3 L2: about the FESOM model
→ indicate the version of the code and maybe a DOI referring to that version

P3 L25: the regional extent could be added on a map with the main features (see a previous comment)

P3 L33: there are other satellites available during that period, why not use them?

P4 L1: The data pre-processing from ERS-2 and Envisat observed ranges to derived DOT heights and follows the descriptions of Müller et al. (2019).
→ it seems a verb is missing

P4 L4: high resolved and with in-situ data combined
→ high resolution and combined with in-situ data

P4 L7-8: The manuscript refers to several of the altimetric variables (SSH, SLA, DOT, DWH, ADT, . . . ) and it would make the reading easier of the mathematical relationships between them was indicated.

P5 Figure 2: Figs 1 and 2 could be merged into a single one where the dots of Fig. 1

are overlayed on Fig. 2 with a common colorbar. Doing so, the comparison would be straightforward and the offset mentioned in the figure caption more obvious.

P5 L8-9: for traceability purpose, indicate the product number from the CMEMS catalog.

P5 L9: same comment concerning the meaning of CMEMS acronym.

P5 L17: must be corrected enabling
→ must be corrected in order to enable

P6 L30: define 'differential water height" (see previous comment concerning the different altimetric variables).

P7 Fig. 3: what is the relation between DOT and DWH?

The colors don't really improve the readability of the flow charts: it seems clear that the altimetric data and the FESOM simulations are the 2 inputs and that blue boxes represent combination (since they receive information from 2 other boxes).

The choice of having uppercase for the variables defined on the grid and lowercase for the along-track variables is maybe not necessary, since you write $coord_{obs}$ or $coords_{sim}$ before the variable name.

The variable names could conserve the same name (with uppercase) but then the spatial coordinates could be X, Y for the grid and x, y for the along-track measurements. Doing so, one avoids the use of $coord_{sim}$ and $coord_{obs}$.

P8 L8-16: for the determination of the number of modes that will be conserved, there is not a real justification. Instead of selecting 50 modes and then computing the resulting RMSE, another approach could be: select the minimal number of modes than ensure that the RMSE is below a given threshold. Figure 4 tends to show that the higher-order modes almost don't contribute to the variance.

P9 Eq. 2: similarly to other comments: the notations can be improved, for instance

avoiding using 3 subscripts ($pc_{comb_i}$). Why not something like $P_i(t)$ instead (for example)?

P9 L21: interpolated to observation coordinates
→ which interpolation method is used?

P10 Fig. 5: this figure could be discarded easily by adding the numbers (1-4) to Figure 3.

P11 L3: which prevents further smoothing effects
→ please explain shortly why the finite element method prevents this.

P11 L5: Comparison with external datasets
→ couldn't you consider a comparison with the altimetric measurements acquired by the satellites not used in this study?

P14 Figure 7: Please note the changed scale of the differences.
→ the changed range?

P15 L15: Whereas the obtained spatial resolution is given by FESOM, characterized by local refinements in ocean current active areas smaller than 1 km.
→ missing verb (or is part of the previous sentence, then should be separated by a comma).

P15 L16: Finite Element mesh
→ sometimes finite element is capitalised, sometimes not, please make uniform.

P17 L6: uncompressed FESOM geostrophic currents
→ define what is the "uncompressed currents".

P17 Data access: the URL relative to ERS-2 points to a news, not to the actual dataset.

p18 Appendix A: this appendix is rather short and could be included directly in the text, hence ensuring the continuity of the calculations.;

[Figure]

**3   Comments on the data files**

Adding the netCDF attributes *standard_name* for the coordinates (longitude, latitude and time) and for the variables. This does not require a full reprocessing but can be done using for instance the nco operator (http://nco.sourceforge.net/), in particular the ncatted command allows one to edit the attributes and can be called as:

```
ncatted -O -h -a standard_name,lon,o,c,longitude 1995_NEGO.nc
ncatted -O -h -a standard_name,lat,o,c,latitude 1995_NEGO.nc
...
```

Concerning the time variable:

The *long_name* is generally specified as "time" and the "units" should be indicate the days (or seconds) since a given date (following the recommendation from the CF conventions: http://cfconventions.org/Data/cf-conventions/cf-conventions-1.7/cf-conventions.html#time-coordinate).

A possible concern is related to the data format: the netCDF is fine and tools are available for reading in many languages, but the finite-element mesh may not be the easiest option for the users, as it may require a regridding. I would suggest to have a similar dataset but provided on a regular grid with a resolution similar to the unstructured grid, or an example (code) of how to go from the unstructured grid to a regular grid.

In the Pangaea preview, the map displays 2 markers labeled 1 and 2, but it is not clear what they are. If it is the spatial coverage, a "rectangle" would be more explicit.

Also in the Pangaea page, it seems the coverage is not exactly the same as in the netCDF for the longitudes:

**Pangaea:** South-bound Latitude: 72.000000 * West-bound Longitude: -29.000000 * North-bound Latitude: 82.000000 * East-bound Longitude: 29.000000

**netCDF:** South-bound Latitude: 72.000000 * West-bound Longitude: -29.803572 * North-bound Latitude: 82.000000 * East-bound Longitude: 29.99896

**Keywords:** the region is specified as "Arctic Ocean" while the article title mentions "the Nordic Seas".

---

## Referee Comment (RC2) · Anonymous Referee #2 · 18 Sep 2019

The paper describes a new dataset of altimetry and associated geostrophic currents, obtained by merging satellite observations and model simulations, and based on principal component analysis. This approach is a novel way to combine both sources of information and seems to produce consistent results.

I note a certain similarity between this approach and the DINEOF method (see e.g. http://modb.oce.ulg.ac.be/mediawiki/index.php/DINEOF_references ) which fills gaps (clouds) in satellite observations using also a PCA method. However in the current paper, the EOFs are obtained from another source of information (a model) than the observations.

[Figure]

The authors have carefully addressed the pre-processing required to merge the 2 datasets (offsets etc).

The paper is well written, generally clear and generally does not contain typos. I only have a few general comments or requests for clarifications. In the remainder of the review, by "dataset" I mean the new product that readers can download.

* page 2 line 7-11. The mentioned regions and main currents should be indicated on a map (e.g. Fig 2 that could then be moved higher in the text).

* page 9 line 12: write out NN. Only on a later page do we learn it means nearest neighbour (page 10 line 1)

* page 9 line 25 : it is not clear why the the description of the "individual steps" is provided separately. For example what is written on page 9 line 27-28, has already been mentioned explicitly before (page 8 line 15). Also the corresponding figure (5) is useless as it is just a zoom on a previous figure

* page 11 line 9-10: "contain" is misleading. Reading this, I could have the impression that the dataset contains the 2 things (satellite & model). By now, the reader has understood that the dataset is build using the 2 sources, but it (i.e. the netcdf file) does not "contain" them. Please rephrase.

* page 11 line 15: the fact that outliers in the results are rejected, has not been mentioned in the method description. We learn only now about it.

* figure 6, right column. Wouldn't a quiver plot (arrows) be more explicit than the colors for indicating the direction ?

* pages 11,13-15 : the authors compare the dataset with different other sources of DOT and currents: "processed" drifter data, original drifter data, ADT. About the second comparison, page 13 lines 16-19, please specify if the rmse computed directly between the drifter and the dataset, is taking as input the original drifter velocity, or is pre-processed (e.g. taking into account only the geostrophic part). Also, line 17, how come

the RMSE is suddenly large (0.13 m/s) in this case, especially compared to the velocity itself ?

Finally, apart from the comparisons proposed in the paper, would it make sense to compare the dataset with the DOT obtained directly from FESOM ? In a perfect world, the dataset would even be compared with a data-assimilating version of FESOM, but I understand this is a whole new work and out of the scope of the article.

* page 17 line 6 : the comparison between "uncompressed" FESOM geostrophic currents and the dataset : what is the meaning of uncompressed ? Also in the article itself, please indicate clearly where you compared the pure FESOM outputs with the dataset, leading up to this phrase in the conclusion.

———————————————————

---

## Referee Comment (RC3) · Anonymous Referee #2 · 18 Sep 2019

The paper describes a new dataset of altimetry and associated geostrophic currents, obtained by merging satellite observations and model simulations, and based on principal component analysis. This approach is a novel way to combine both sources of information and seems to produce consistent results.

I note a certain similarity between this approach and the DINEOF method (see e.g. http://modb.oce.ulg.ac.be/mediawiki/index.php/DINEOF_references ) which fills gaps (clouds) in satellite observations using also a PCA method. However in the current paper, the EOFs are obtained from another source of information (a model) than the observations.

[Figure]

The authors have carefully addressed the pre-processing required to merge the 2 datasets (offsets etc).

The dataset itself is easy to download and read (at least for users with experience of netcdf files, which is standard). Given each file is large (∼2 GB), users who wish to download the entire dataset would maybe appreciate compressed netcdf files. Most services are switching to netcdf4 compressed files (e.g. CMEMS).

The paper is well written, generally clear and generally does not contain typos. I only have a few general comments or requests for clarifications. In the remainder of the review, by "dataset" I mean the new product that readers can download.

* page 2 line 7-11. The mentioned regions and main currents should be indicated on a map (e.g. Fig 2 that could then be moved higher in the text).

* page 9 line 12: write out NN. Only on a later page do we learn it means nearest neighbour (page 10 line 1)

* page 9 line 25 : it is not clear why the the description of the "individual steps" is provided separately. For example what is written on page 9 line 27-28, has already been mentioned explicitly before (page 8 line 15). Also the corresponding figure (5) is useless as it is just a zoom on a previous figure

* page 11 line 9-10: "contain" is misleading. Reading this, I could have the impression that the dataset contains the 2 things (satellite & model). By now, the reader has understood that the dataset is build using the 2 sources, but it (i.e. the netcdf file) does not "contain" them. Please rephrase.

* page 11 line 15: the fact that outliers in the results are rejected, has not been mentioned in the method description. We learn only now about it.

* figure 6, right column. Wouldn't a quiver plot (arrows) be more explicit than the colors for indicating the direction ?

* pages 11,13-15 : the authors compare the dataset with different other sources of DOT and currents: "processed" drifter data, original drifter data, ADT. About the second comparison, page 13 lines 16-19, please specify if the rmse computed directly between the drifter and the dataset, is taking as input the original drifter velocity, or is pre-processed (e.g. taking into account only the geostrophic part). Also, line 17, how come the RMSE is suddenly large (0.13 m/s) in this case, especially compared to the velocity itself ?

Finally, apart from the comparisons proposed in the paper, would it make sense to compare the dataset with the DOT obtained directly from FESOM ? In a perfect world, the dataset would even be compared with a data-assimilating version of FESOM, but I understand this is a whole new work and out of the scope of the article.

* page 17 line 6 : the comparison between "uncompressed" FESOM geostrophic currents and the dataset : what is the meaning of uncompressed ? Also in the article itself, please indicate clearly where you compared the pure FESOM outputs with the dataset, leading up to this phrase in the conclusion.

---

## Author Comment (AC1) · 15 Oct 2019

We thank the Reviewer for the careful and constructive comments. The suggestions and corrections have greatly improved the quality of this manuscript.

1 General comments

The article describe a new dataset constructed using altimetry, along-track measurements with the outputs a high-resolution numerical model in the "northern Nordic Seas". Overall the article is clearly written and all the details concerning the processing is provided. Nevertheless I formulate 3 main comments regarding the present version of the manuscript:

1. The innovation: the manuscript would be enriched if the authors add a brief justification of the advantages of their new dataset with respect to results of numerical simulations in the same area.

When combining observation data with model simulations we combine and capitalize the advantages of both datasets and get rid of their individual disadvantages. Numerical simulations (without data assimilation) rely on the underlying mathematical/physical relations, which are in all cases simplifications of reality based on specific assumptions. In contrast, observations capture the reality. However, they suffer from measurement errors and sparse spatial resolution. In our study, we mainly rely on the observational side and fill data gaps by model simulations. It's not our intention to compete with ocean models, but to improve altimetry derived datasets. In the manuscript, we showed that the combined dataset has a lot of advantages with respect to satellite only solutions (see comparison with ADT).

We theoretically discuss the differences between models and observations in the introduction, especially in line 10 to 18 on page 3. We add some more information in the introduction on this. A performance comparison between FESOM and combined data is not in the focus of the paper.

We added the following text passage to the manuscript P2 L31:

„Besides observation-based ocean circulation products, model simulations provide information about the ocean dynamics. ***In general, their resolution is much better than these of observations, however, they rely on the underlying mathematical or physical formulations, which naturally contain simplifications and suffer from deficiencies in process descriptions.*** Ocean models differ in spatio-temporal resolutions, forcing model background and underlying mathematical formulations. Recent developments are focusing on so-called unstructured ocean models…"

2. The dataset itself: while it is clear that the FESOM model is based on finite elements, having the different variables on such an unstructured grid may not be the easiest option for most of the users.

We agree with the Reviewer and added a python code to the datastorage (PANGAEA) for an easy regridding of the combined dataset. Please also see section 3 "Comments on data files".

3. The notations: starting from the schema (Figure 3), the way the different variables are defined make the developments difficult to follow. Suggestions are provided in the specific comments. I believe that the overall quality and the readability would be improved if the authors can address these 3 points.

We agree with the Reviewer and changed the naming of the variables. Please see section 2 "Specific comments".

2 Specific comments

P1 L9: it is stated that the presented method differs from data assimilation because it substitutes altimetry data with the model output; however it is not clear in the manuscript what is the advantage over the dataset obtained with a pure assimilation approach.

Data assimilation is a completely different combination strategy. Of course, it would be interesting to compare both combination approaches, however, no assimilated FESOM version is available for such comparison, and it is beyond the scope of this paper, to create such a model. Moreover, we do not claim that our approach outperforms an assimilation approach. Our product is mainly focused on the observational side by filling in modeled DOT elevations, where altimetry data is missing or corrupted. This is already stated in the manuscript (page 3, line 14/15).

P1 L14: the by altimetry obtained annual signal→the annual signal obtained by altimetry

We changed the text, accordingly.

P1 L17: and the temporal variability of the altimetry along-track derived DOT heights→specify the temporal resolution (1 day according to the netCDF)

In fact, the temporal resolution of satellite altimetry depends on the repeat cycle of the used mission. What is meant is the high along-track (spatial) resolution (20 Hz measurements). Thanks for pointing out that this sentence is misleading. We replaced it by:
The resulting final product exploits the advantages of both input data sets combining the accurate high-frequent along-track altimetry observations with the high spatial resolution of the ocean model. It is provided with daily resolution on the original FESOM grid.

P1 L21: what justifies the difference in the order of magnitude (0.004 vs 0.02 m/s) for the zonal and meridional components?

The study area is characterized by currents with a predominantly meridional flow direction (e.g. East Greenland Current, West Spitsbergen Current). Due to this reason,

the meridional signal shows a stronger magnitude and variance resulting in bigger differences in the v-component (north-south direction). Please also find P 13 Line 13 and the new added map (Figure 1) showing the major current systems in the investigation area.

P2 L1: drifter location interpolated combined geostrophic velocity components→combined geostrophic velocity components interpolated onto the drifter locations

We changed the text, accordingly.

P2 L2: a general map with the main geographical features would help the readers unfamiliar with the region of interest.

We added a map including the major bathymetric features and major current systems.

[Figure]

Bathymetry of the study area (norhern Nordic Seas, Fram Strait) based on RTopo2 topography model (Schaffer et al. 2016). Major current systems (West-Spitsbergen Current, WSC; East Greenland Current, EGC) are displayed by arrows in red (inflowing Atlantic water) and blue (returning polar water). Contour lines indicate depths of -450 and -1500 meters.

P2 L18: by altimetry derived geostrophic ocean currents→geostrophic ocean currents derived by altimetry or altimetry-derived geostrophic ocean currents

We changed the text, accordingly.

*[...] **geostrophic ocean currents derived by altimetry** suffer from [...]*

P2 L29: CMEMS stands for "Copernicus Marine Environment Monitoring Service"

We apologize for the wrong abbreviation and change the text, accordingly

P2 L32: and underlying mathematical functions→mathematical formulations

We changed the text, accordingly.

P2 L31: information of the ocean dynamics→information on/about the ocean dynamics

We changed the text, accordingly.

P3 L2: about the FESOM model→indicate the version of the code and maybe a DOI referring to that version

We added some information to the text. Please also note section 7 "Data availability", where we provide a data link including a DOI and a link to the model code. For an easier readability, we use the abbreviation FESOM instead of FESOMv1.4.

New text: *Recent developments* **focus** *on so-called unstructured ocean models, allowing for locally highly refined spatial resolutions (Danilov, (2013)), while keeping a coarser resolution in other regions of the Earth* **(e.g. Finite Element Sea ice Ocean Model, Wang et al, (2014) or Model for Prediction Across Scale Ocean model (MPAS-Ocean), Ringler et al. (2013)). One of the unstructured models is the Finite Element Sea ice Ocean Model version 1.4 (FESOMv1.4) described by Wang et al. (2014)). Please note, in the following text FESOMv1.4 is abbreviated by FESOM.**

P3 L25: the regional extent could be added on a map with the main features (see previous comment)

Please see above, we added a map to the article.

P3 L33: there are other satellites available during that period, why not use them?

In the present article, we use only radar altimetry missions completely covering the investigation area. This prevents the usage of CNES/NASA missions (TOPEX/Poseidon, Jason-1, 2) and NOAA mission GEOSAT Follow-On showing an orbit inclination of about 66 degrees and ~72 degrees. Furthermore, the altimetry data preprocessing (i.e the unsupervised classification approach and the radar waveform retracking algorithm, ALES+) is only applicable to pulse-limited satellite altimetry missions. This limits the available missions to ERS and Envisat. However, in future it should be possible to extend the classification and retracking approach to Delay-Doppler (SAR) altimetry missions (CryoSat-2, Sentinel-3A/B). Please see also chapter 6, Summary and Conclusions.

P4 L1: The data pre-processing from ERS-2 and Envisat observed ranges to derived DOT heights and follows the descriptions of Müller et al. (2019).→it seems a verb is missing

We follow the Reviewer's comment and changed the sentence. It seems the „and" is at a wrong place.

*The data pre-processing from ERS-2 and Envisat observed ranges to derived DOT heights  follows the descriptions of Müller et al. (2019).*

P4 L4: high resolved and with in-situ data combined→high resolution and combined with in-situ data

We changed the text, accordingly.

P4 L7-8: The manuscript refers to several of the altimetric variables (SSH, SLA, DOT,DWH, ADT, . . . ) and it would make the reading easier of the mathematical relationships between them was indicated.

We follow the Reviewer's suggestion and added a list of abbreviations and nomenclature of altimetry as well as FESOM related variables. Please see Appendix A.

P5 Figure 2: Figs 1 and 2 could be merged into a single one where the dots of Fig. 1 are overlayed on Fig. 2 with a common colorbar. Doing so, the comparison would be straightforward and the offset mentioned in the figure caption more obvious.

Figure 1 and Figure 2 should just show the input datasets (altimetry and model) before the combination process. The main point here is the different spatial data distribution. The offset, which has already discussed in detail in Müller et al. 2019 is only of minor interest (as it is reduced before the combination). We think the combination of both plots would make it difficult to see the satellite tracks, since the FESOM representation will dominate the along-track data of altimetry. We decided to change the caption of Figure 2.

Note the offset in comparison to Figure 1 → Note the different scaling of colorbars in comparison to Figure 1.

P5 L8-9: for traceability purpose, indicate the product number from the CMEMS catalog.

We added the product number to the references.

P5 L9: same comment concerning the meaning of CMEMS acronym.
We thank the Reviewer and changed the text, accordingly.

P5 L17: must be corrected enabling→must be corrected in order to enable

We changed the text, accordingly.

P6 L30: define 'differential water height" (see previous comment concerning the different altimetric variables).

Please see Appendix A and previous comment.

P7 Fig. 3: what is the relation between DOT and DWH?

Please see chapter 2.2 and for further explanations the publication of Androsov et al., 2018. DWH are differential water heights with respect to a virtual defined reference surface. In case of FESOM DWH are very similar to DOT heights, however secular changes like gravitational forces or the Glacial Isostatic Adjustment (GIA) are not considered. Following the explanations of Androsov et al. (2018) the volume conservation of FESOM leads to correct modeled horizontal gradients but to a constant offset (~47cm) between the simulated heights and the geodetic by altimetry derived DOT. Moreover, the offset can be also confirmed in Müller et al. (2019).

Androsov, A., Nerger, L., Schnur, R., Schröter, J., Albertella, A., Rummel, R., Savcenko, R., Bosch, W., Skachko, S., and Danilov, S.: On the assimilation of absolute geodetic dynamic topography in a global ocean model: impact on the deep ocean state, Journal of Geodesy, https://doi.org/10.1007/s00190-018-1151-1, 2018.

Müller, F. L., Wekerle, C., Dettmering, D., Passaro, M., Bosch, W., and Seitz, F.: Dynamic ocean topography of the northern Nordic seas: a comparison between satellite altimetry and ocean modeling, The Cryosphere, 13, 611–626, https://doi.org/10.5194/tc-13-611-2019, 2019

The colors don't really improve the readability of the flow charts: it seems clear that the altimetric data and the FESOM simulations are the 2 inputs and that blue boxes represent combination (since they receive information from 2 other boxes).

We decided to keep the different colors. Even if their definition might be obvious, their usage will not degrade the readability of the plot. However, we agree to unify the colors of the input datasets. Plot and caption have been updated.

The choice of having uppercase for the variables defined on the grid and lowercase for the along-track variables is maybe not necessary, since you write coord$_{obs}$ or coords$_{sim}$ before the variable name.
The variable names could conserve the same name (with uppercase) but then the spatial coordinates could be X, Y for the grid and x, y for the along-track measurements. Doing so, one avoids the use of coord$_{sim}$ and coord$_{obs}$.

We follow the Reviewer's suggestions and change the labeling of the various coordinate representations. Please also note the changed caption of Figure 3 (processing scheme).

P8 L8-16: for the determination of the number of modes that will be conserved, there is not a real justification. Instead of selecting 50 modes and then computing the resulting RMSE, another approach could be: select the minimal number of modes than ensure that the RMSE is below a given threshold. Figure 4 tends to show that the higher-order modes almost don't contribute to the variance.

We agree with the Reviewer's opinion. Maybe the sentence is misleading. Our goal is to ensure a 10 mm error budget in the pure reconstruction of FESOM. Thus, we tested how many modes are needed to reach this 10 mm error in maximum. The Reviewer is absolutely correct by explaining that higher-order Modes don't contribute to the total variance. We also wanted to show this in Figure 4.

We changed the sentence: **It is decided to use a RMSE threshold of 10mm, corresponding to 50 Modes and a summed variance of more than 99%.**

P9 Eq. 2: similarly to other comments: the notations can be improved, for instance avoiding using 3 subscripts (pc$_{combi}$). Why not something like P$_i$(t)instead (for example)?

We follow the Reviewer's suggestions and change the labeling. Please also note the changed caption of Figure 3 (processing scheme).

P9 L21: interpolated to observation coordinates→which interpolation method is used?

We used a simple nearest neighbor interpolation. We added further information to the text.

*The individual weights are defined by using external sea ice concentration from the National Snow and Ice Data Center (NSIDC, Fetterer et al. (2017)) interpolated v**ia nearest-neighbor interpolation** to the observation coordinates **considering** an [...]*

P10 Fig. 5: this figure could be discarded easily by adding the numbers (1-4) to Figure3.

We agree with the Reviewer since Figure 5 is a zoom of Figure 3, however, we want to highlight the key stages (1-4) of the combination process. We think it's easier to understand for readers, who are no experts in PCA to follow the different processing stages step by step.

P11 L3: which prevents further smoothing effects→please explain shortly why the finite element method prevents this.

The benefit of computing geostrophic currents via finite element methods can be seen in a direct derivation on the grid nodes without an additional interpolation. Normally, an inclined plane is fitted to the grid nodes computing zonal and meridional slope coefficients. Therefore, neighboring height values must be interpolated using a certain cap size, which causes smoothing and damping effects.

We changed the text to: "… which prevents further smoothing effects, **since no re-gridding to a regular grid is necessary**."

P11 L5: Comparison with external datasets→couldn't you consider a comparison with the altimetric measurements acquired by the satellites not used in this study?

We agree with the Reviewer that it is a good idea to use altimetry data, which is not part of the study as validation data. However, currently, there is no altimetry data we can use, which is consistent with the observations used for the product generation. Our algorithms are valid for pulse-limited, i.e. conventional satellite altimetry missions. This holds for the  classification approach as well as for the retracking of the radar echoes. Delay-Doppler missions, such as CryoSat-2 are based on another observing principle resulting in different radar echoes. ALES+, the physical retracker used in the present study is not applicable to Delay-Doppler data. Thus, a fair and consistent comparison or validation is currently not possible with this mission.

Moreover, radar altimetry data from CNES/NASA missions TOPEX/Poseidon, Jason-1,2, 3 and NOAA mission GEOSAT Follow-On (GFO) can't be used due their limited spatial coverage and orbit characteristics (~66°N/S for Jason-1,2,3 and ~72°N/S for GFO).

In future, it's planned to adapt ALES+ to Delay-Doppler radar echoes in order to include further observed DOT heights to the combination process.

P14 Figure 7: Please note the changed scale of the differences.→the changed range?

We apologize for this wrong caption. There is no changed range. Thank you very much for the comment.

P15 L15: Whereas the obtained spatial resolution is given by FESOM, characterizedby local refinements in ocean current active areas smaller than 1 km.→missing verb (or is part of the previous sentence, then should be separated by a comma).

We changed the sentence to: "the obtained spatial resolution is *defined* by FESOM, *which is* characterized by local refinements…."

P15 L16: Finite Element mesh→sometimes finite element is capitalised, sometimes not, please make uniform.

Thank you for the hint. We uniformed the text (using finite element).

P17 L6: uncompressed FESOM geostrophic currents→define what is the "uncompressed currents".

This sentence was a fragment in the summary/conclusion from a previous version of the article, sorry. The sentence has been deleted now, thank you for the comment.

P17 Data access: the URL relative to ERS-2 points to a news, not to the actual dataset.

We apologize for the wrong link and updated the URL.

New: https://earth.esa.int/web/guest/-/radar-altimeter-reaper-sensor-geophysical-data-record-sgdr

p18 Appendix A: this appendix is rather short and could be included directly in the text,hence ensuring the continuity of the calculations

We thought about this comment and decided to erase Appendix A, because it's a standard procedure and does not contain any special mathematical formulation. More information regarding least-squares can be also found in Koch (1999), which is a standard reference for parameter estimation.

3 Comments on the data files

We really thank the Reviewer for the information about the function "ncatted" and the comments.

Adding the netCDF attributes standard_name for the coordinates (longitude, latitude and time) and for the variables. This does not require a full reprocessing but can be done using for instance the nco operator (http://nco.sourceforge.net/), in particular then catted command allows one to edit the attributes and can be called as:

```
ncatted -O -h -a standard_name,lon,o,c,longitude 1995_NEGO.nc
ncatted -O -h -a standard_name,lat,o,c,latitude 1995_NEGO.nc
```

Concerning the time variable: The long_name is generally specified as "time" and the "units" should be indicate the days (or seconds) since a given date (following the recommendation from the CF conventions: http://cfconventions.org/Data/cf-conventions/cf-conventions-1.7/cf-conventions.html#time-coordinate).

We added "standard_name" to the NetCDF files.

A possible concern is related to the data format: the netCDF is fine and tools are available for reading in many languages, but the finite-element mesh may not be the easiest option for the users, as it may require a regridding. I would suggest to have a similar dataset but provided on a regular grid with a resolution similar to the unstructured grid, or an example (code) of how to go from the unstructured grid to a regular grid.

We decided to provide an example python code, using The Generic Mapping Tools (GMT, http://gmt.soest.hawaii.edu/projects/gmt), for an easy transformation to a regular grid. GMT is a commonly used, well validated open source software and can read directly NetCDF files. A short function description is added to the user manual.

In the Pangaea preview, the map displays 2 markers labeled 1 and 2, but it is not clear what they are. If it is the spatial coverage, a "rectangle" would be more explicit. Also in the Pangaea page, it seems the coverage is not exactly the same as in the netCDF for the longitudes:

Pangaea: South-bound Latitude: 72.000000 * West-bound Longitude: -29.000000 *North-bound Latitude: 82.000000 * East-bound Longitude: 29.000000
netCDF:South-bound Latitude: 72.000000 * West-bound Longitude: -29.803572 *North-bound Latitude: 82.000000 * East-bound Longitude: 29.99896

In case of the overview map we contacted PANGAEA. Unfortunately, it's not possible to add a rectangle for highlighting the study area. We know that a simple Mercator (Google) projection is not the best for polar regions, however PANGAEA does not have any

alternatives. However, the coordinates should be right. We sent a notice to PANGAEA to correct this.

Keywords:the region is specified as "Arctic Ocean" while the article title mentions "the Nordic Seas"

The Keywords are adapted.

---

## Author Comment (AC2) · 15 Oct 2019

We thank the Reviewer for the careful and constructive comments. The suggestions and corrections have greatly improved the quality of this manuscript.

The paper describes a new dataset of altimetry and associated geostrophic currents, obtained by merging satellite observations and model simulations, and based on principal component analysis. This approach is a novel way to combine both sources of information and seems to produce consistent results.

I note a certain similarity between this approach and the DINEOF method (see e.g.http://modb.oce.ulg.ac.be/mediawiki/index.php/DINEOF_references ) which fills gaps(clouds) in satellite observations using also a PCA method. However in the current paper, the EOFs are obtained from another source of information (a model) than the observations.

Thank you for this link. We were not aware of this method. I fact, this is an interesting method, which is based on EOF as our approach is. However, one big difference is, that we do not use the altimetry observations for EOF analysis, since they are not available in fixed locations and with uniform epochs. In contrast, we use external data (a model) in order to fill the data gaps. That's why we decided not to mention DINEOF in the introduction of our manuscript.

The authors have carefully addressed the preprocessing required to merge the 2 datasets (offsets etc). The dataset itself is easy to download and read (at least for users with experience of netcdf files, which is standard). Given each file is large (~2 GB), users who wish to download the entire dataset would maybe appreciate compressed netcdf files. Most services are switching to netcdf4 compressed files (e.g. CMEMS).

We follow the Reviewer's suggestion and compress the NetCDF files with the highest compression level, which results in a total reduction of about 14GB.

The paper is well written, generally clear and generally does not contain typos. I only have a few general comments or requests for clarifications. In the remainder of the review, by "dataset" I mean the new product that readers can download.

* page 2 line 7-11. The mentioned regions and main currents should be indicated on a map (e.g. Fig 2 that could then be moved higher in the text).

We follow the Reviewer's suggestion and added a map containing major currents and bathymetric features to the introduction (see new Figure 1)

[Figure]

*Bathymetry of the study area (norhern Nordic Seas, Fram Strait) based on RTopo2 topography model (Schaffer et al. 2016). Major current systems (West-Spitsbergen Current, WSC; East Greenland Current, EGC) are displayed by arrows in red (inflowing Atlantic water) and blue (returning polar water). Contour lines indicate depths of -450 and -1500 meters.*

* page 9 line 12: write out NN. Only on a later page do we learn it means nearest neighbour (page 10 line 1)

We changed the text, accordingly.

* page 9 line 25 : it is not clear why the the description of the "individual steps" is provided separately. For example what is written on page 9 line 27-28, has already been mentioned explicitly before (page 8 line 15). Also the corresponding figure (5) is useless as it is just a zoom on a previous figure

You are right that the list of individual combination steps is redundant information. The intention is to highlight the key processing steps in order to make it easy for the readers to understand the procedure. We think that the Figure is helpful for other, less experienced readers and decided to keep the figure and explanations.

* page 11 line 9-10: "contain" is misleading. Reading this, I could have the impression that the dataset contains the 2 things (satellite & model). By now, the reader has understood that the dataset is build using the 2 sources, but it (i.e. the netcdf file) does not "contain" them. Please rephrase.

We agree and changed the text, accordingly.

*The combined DOT and geostrophic current velocity fields are based on DOT heights from satellite altimetry [...].*

\* page 11 line 15: the fact that outliers in the results are rejected, has not been mentioned in the method description. We learn only now about it.

We agree and added some text to section 3.2. (step 4). Text passages in chapter 4 are now redundant and deleted.

New text chapter 3.2 (P 10 Line 5) :

*Furthermore, an outlier detection based on an accuracy determination of the combined principal components is performed to reject erroneous combination estimations.*

Deleted text chapter 4: (P 11 Line 15):

*Furthermore, an outlier detection based on the accuracy of the computed combined principal components is performed rejecting erroneous combination estimations.*

\* figure 6, right column. Wouldn't a quiver plot (arrows) be more explicit than the colors for indicating the direction?

We tried quiver plots for displaying the flow direction. However, the unstructured grid is very dense (~1km), resulting in a lot of arrows reducing the readability. Therefore we decided to indicate the flow direction by using a circular phase map ranging from [-180° to 180°].

\* pages 11,13-15 : the authors compare the dataset with different other sources of DOT and currents: "processed" drifter data, original drifter data, ADT. About the second comparison, page 13 lines 16-19, please specify if the rmse computed directly between the drifter and the dataset, is taking as input the original drifter velocity, or is preprocessed (e.g. taking into account only the geostrophic part). Also, line 17, how come the RMSE is suddenly large (0.13 m/s) in this case, especially compared to the velocity itself?

We apologize for the confusion. We only use "processed" drifter data, which means only the geostrophic part of the drifter total velocity. We use a 6 hourly interpolated drifter dataset and reduce the individual observations by a-geostrophic (wind, wave) movements. In this second comparison, we directly compare the geostrophic drifter trajectories, which means the single geostrophic drifter observations, with the combination velocities. The numbers are higher than for the gridded comparison, because the individual drifter trajectories are characterized by a high noise budget and a strong variability.

Change text to P13 L16: *When computing the RMSE between the measured geostrophic velocities and the combined velocities based on the individual trajectories for each drifter, a mean of...*

Finally, apart from the comparisons proposed in the paper, would it make sense to compare the dataset with the DOT obtained directly from FESOM? In a perfect world, the

dataset would even be compared with a data-assimilating version of FESOM, but I understand this is a whole new work and out of the scope of the article.

We don't think that a comparison of the new product with the original FESOM model is able to provide new findings. Of course, differences will exist, but the observations will not improve the model (for this purpose a data assimilation or calibration would be required, which keeps the model physics). This was not the aim of the study. We want to improve observation-based velocities, that's why we only include comparisons to observation-only products.

* page 17 line 6 : the comparison between "uncompressed" FESOM geostrophic cur-rents and the dataset : what is the meaning of uncompressed? Also in the article itself, please indicate clearly where you compared the pure FESOM outputs with the dataset, leading up to this phrase in the conclusion.

This sentence has been deleted, since it was wrong. There is no comparison between FESOM and the combined dataset.